# Machine learning uncovers the *Pseudomonas syringae* transcriptome in microbial communities and during infection

Heera Bajpe,[1] Kevin Rychel,[1] Cameron R. Lamoureux,[1] Anand V. Sastry,[1] Bernhard O. Palsson[1,2,3,4,5]

**ABSTRACT**  The transcriptional regulatory network (TRN) of the phytopathogen *Pseudomonas syringae* pv. *tomato* DC3000 regulates its response to environmental stimuli, including interactions with hosts and neighboring bacteria. Despite the importance of transcriptional regulation during these agriculturally significant interactions, a comprehensive understanding of the TRN of *P. syringae* is yet to be achieved. Here, we collected and decomposed a compendium of public RNA-seq data from *P. syringae* to obtain 45 independently modulated gene sets (iModulons) that quantitatively describe the TRN and its activity state across diverse conditions. Through iModulon analysis, we (i) untangle the complex interspecies interactions between *P. syringae* and other terrestrial bacteria in cocultures, (ii) expand the current understanding of the *Arabidopsis thaliana-P. syringae* interaction, and (iii) elucidate the AlgU-dependent regulation of flagellar gene expression. The modularized TRN yields a unique understanding of interaction-specific transcriptional regulation in *P. syringae*.

**IMPORTANCE**  *Pseudomonas syringae* pv. *tomato* DC3000 is a model plant pathogen that infects tomatoes and *Arabidopsis thaliana*. The current understanding of global transcriptional regulation in the pathogen is limited. Here, we applied iModulon analysis to a compendium of RNA-seq data to unravel its transcriptional regulatory network. We characterize each co-regulated gene set, revealing the activity of major regulators across diverse conditions. We provide new insights on the transcriptional dynamics in interactions with the plant immune system and with other bacterial species, such as AlgU-dependent regulation of flagellar genes during plant infection and downregulation of siderophore production in the presence of a siderophore cheater. This study demonstrates the novel application of iModulons in studying temporal dynamics during host-pathogen and microbe-microbe interactions, and reveals specific insights of interest.

**KEYWORDS**  *Pseudomonas syringae*, independent component analysis, transcriptomics, gene regulation, data mining, microbial interactions

<P>*seudomonas syringae* pv. *tomato* DC3000 is a gram-negative phytopathogen that infects tomato plants, causing major crop losses worldwide (1). *P. syringae* is also extensively studied as a model for investigating plant-pathogen interactions in *Arabidopsis thaliana*. Despite its significance as a foliar plant pathogen, our knowledge of the impact of plant immunity and interactions with neighboring bacterial species in the environment on transcriptional regulation in *P. syringae* is limited. Additionally, transcriptional regulation in *P. syringae* is poorly understood in comparison to more widely studied relatives such as *Pseudomonas aeruginosa* and *Pseudomonas putida*. Given the immense agricultural impact of this pathogen, a deeper mechanistic understanding of the transcriptional regulatory network (TRN) of *P. syringae* is required. Understanding</P>

Address correspondence to Bernhard O. Palsson, palsson@ucsd.edu.

The authors declare no conflict of interest.

the TRN is imperative to improving our knowledge of the fundamentals of its pathogenesis and survival strategies at a systems level.

Independent component analysis (ICA) was developed to identify statistically independent components from complex data sets (2). Specifically, the application of ICA to compendia of bacterial RNA sequencing (RNA-seq) data has been found to be successful in extracting independently modulated sets of genes, called iModulons (3). In addition to its gene composition, ICA determines the activities of each iModulon across all growth conditions found in the RNA-seq compendium (3). An identified iModulon consists of a set of genes that is often under the regulation of a specific transcriptional regulator. iModulons thus represent clear and interpretable biological signals.

With the number of iModulons being far lower than the number of genes, the use of iModulons reduces the complexity of studying the TRN greatly, while providing a global, systems-wide perspective of organism function. We have previously employed this top-down approach to unravel the TRN of several bacterial species, including *E. coli* (3, 4), *Bacillus subtilis* (5), *P. aeruginosa* (6, 7), and *P. putida* (8). The iModulon structures of these, as well as other bacterial species, can be found at iModulonDB.org (9). Through these studies, we have been able to gain deep insights into the functions of the TRN of these organisms, such as better definition of known regulons, characterization of new regulons and previously unknown genes, and identification of evolutionary transcriptomic trade-offs.

In the present study, we performed ICA decomposition on a compendium of RNA-seq data consisting of all high quality publicly available data for *P. syringae*. A total of 45 iModulons were extracted from 202 RNA-seq profiles, capturing 64% of the variance in the data set. Each iModulon was curated with regulator, function, and category annotations. Through our analysis, we uncover novel signals comprising the transcriptional responses of *P. syringae* in relevant conditions, and further advance the utility of iModulons as a tool for hypothesis generation. Additionally, we demonstrate the use of iModulons in interpreting complex bacterial interactions in microbial communities, as well as responses during plant-pathogen interactions. Specifically, we studied the dynamics of transcriptional regulation in *P. syringae* cocultured with other terrestrial bacterial species (*Burkholderia thailandensis* and *Chromobacterium violaceum*) to understand competition and survival strategies of the pathogen in the presence of neighbors it often coexists with in natural habitats (10, 11). Additionally, we analyzed the dynamic changes that the TRN of *P. syringae* undergoes as a result of interacting with the *A. thaliana* immune system, adding to existing knowledge on the well-studied plant-pathogen interaction (12, 13). Finally, we utilized iModulons to reveal finer details in the downregulation of flagellar genes in the apoplast mediated by AlgU (14). All of the data and curated iModulons from this study are available to browse, search, and download on iModulonDB.org.

## RESULTS

### ICA finds 45 iModulons in *P. syringae* transcriptomic data

We collected all public data available for *P. syringae* on the Sequence Read Archives (SRA) database of NCBI using our previously established pipeline (see Materials and Methods) (15). These data were collected from studies investigating plant-pathogen interactions, including a range of plant immunity-inducing conditions and bacterial mutations, and cocultures of *P. syringae* with bacterial species found in terrestrial environments. These samples have been briefly described in Table 1, with a more detailed description included in Data S1. Overall, the data set consisted of a total of 202 samples obtained under 56 unique conditions across six projects, five of which were published studies (10–14, 16). For each project, a baseline condition was selected and all data from the project were centered to the mean of the condition, to reduce batch effects (Table S1). We then applied our ICA algorithm to the full compendium (see Materials and Methods; Fig. 1a and b) (17). ICA extracted 45 iModulons that together explain 64% of the variance in gene expression. Each iModulon represents the effect of transcription factors under

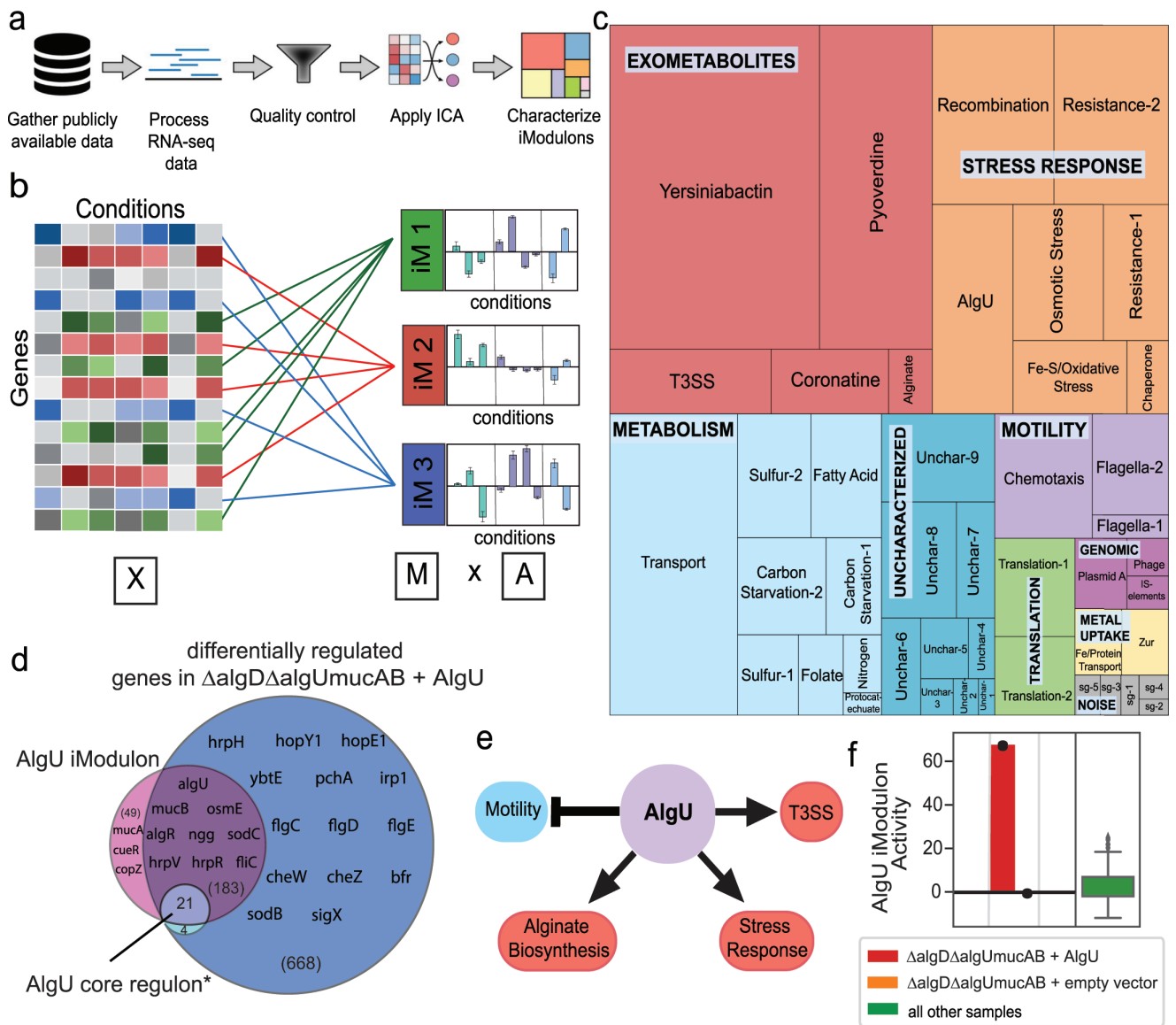

FIG 1  ICA decomposition of the *P. syringae* pv. *tomato* DC3000 RNA-seq data compendium. (a) Data processing pipeline for running ICA on publicly available data to generate iModulons for *P. syringae*. This diagram was adapted from Anand et al. (17). (b) ICA decomposed the gene expression profiles (X; 5,906 genes by 202 conditions) into an "M" matrix containing the weights of each gene in the 45 iModulons identified and an "A" matrix containing the activity of each iModulon across all conditions (iM: iModulon). This figure was created using BioRender (BioRender.com). (c) A treemap of the 45 iModulons of *P. syringae*, with the size of each rectangle representing the percentage of variance in the data each iModulon explains. (d) Venn diagram of genes in the AlgU iModulon, enriched for the transcription factor AlgU, genes differentially expressed in ΔalgDΔalgUmucAB mutants transformed with an AlgU expression vector compared to mutants transformed with an empty vector, and genes in the AlgU core regulon defined based on genes regulated by AlgU in *P. syringae* pv. *syringae* B728 and *P. aeruginosa* PAO1 (14). (e) The AlgU iModulon contains genes associated with motility, alginate biosynthesis, stress responses, and the type III secretion system (T3SS), all of which are known to be regulated by AlgU (18). (f) Activity levels of the AlgU iModulon in AlgU expression strains, in mutants transformed with an empty vector, and all other samples (*n* = 196).

various conditions, and thus explained variance is not only statistical, as in principal component analysis (PCA), but also reflects real biological mechanisms.

After characterization, the 45 iModulons were classified into four broad categories: functional, genomic, uncharacterized, and noise. The iModulons classified as "Functional" were further divided into six subcategories: "Exometabolites," "Metabolism," "Stress response," "Motility," "Translation," and "Metal uptake" (Fig. 1c). Due to a lack of direct evidence of regulation by a transcription factor, these iModulons were named based

**TABLE 1** Experimental context of analyzed data sets[a,b]

| Reference | Condition | Bacterial strains | Host genotype | Infiltration agents |
|---|---|---|---|---|
| (14) | *In vitro* | Pto Δ*algD*Δ*algUmucAB* and Pto Δ*algD*Δ*algUmucAB* + AlgU | Not applicable | None |
| (10, 11) | *In vitro*, coculture (monoculture, two- and three-member cocultures) | Pto, *C. violaceum,* and *B. thailandensis* | Not applicable | None |
| (16) | *In vivo* | Pto | *A. thaliana pad4* and *sid2* mutants | None |
| (12) | *In vitro*, growth in minimal media | Pto | Not applicable | None |
| (12) | *In vivo* | Pto and Pto ectopically expressing AvrRpt2 | *A. thaliana* wild-type Col-0 and *pad4, sid2, npr1, dde2, ein2, pmr4, cyp79b2, cyp79b3, stp1, stp13, rps2,* and *rpm1* mutants | Water, chitin, salicylic acid |
| (12) | *In vivo* | Pto | *A. thaliana* wild-type Col-0 | flg22, DMSO |
| (13) | *In vivo* | Pto | *A. thaliana* wild-type Col-0 | flg22, DMSO |

[a]Pto: *P. syringae* pv. *tomato* DC3000.
[b]See Supplementary Data S1 for details of each sample.

on the cellular function that they carry out. Of the 28 "Functional" iModulons, the "Exometabolites" subcategory, which includes siderophores and virulence functions, explains the highest percentage of variance in the data set (20.74%). Other major functional categories include "Metabolism" (13.55%), "Stress response" (15.19%), and "Motility" (3.77%). Additionally, "Genomic" iModulons (1.09%) contain genes that were likely differentially expressed due to genetic perturbations such as plasmid manipulation. The "Uncharacterized" iModulons (3.8%) represent groups of genes with unknown functions or unclear activating conditions. Finally, the "Noise" iModulons (0.62%) each contain a single gene, which is likely the result of noise in the expression of those genes.

The usefulness of iModulons is illustrated through analysis of the AlgU iModulon. One of the studies included in our compendium investigated genes differentially expressed due to the overexpression of AlgU using traditional statistical methods. It was found that genes involved in motility were differentially downregulated, while genes associated with alginate biosynthesis, stress responses, and the type III secretion system (T3SS) were upregulated. Although AlgU is known to regulate genes in these functional categories, the differential regulation of these genes may be due to indirect transcription factor interactions (Fig. 1e) (18). We found an iModulon, which we named the AlgU iModulon, that represents a fraction of these differentially expressed genes, while representing all the above-mentioned functional categories (Fig. 1d). Additionally, a majority of the genes in the AlgU core regulon, defined as genes known to be regulated by AlgU in *P. syringae* pv. *syringae* B728 and *P. aeruginosa* PAO1, are members of the AlgU iModulon (14). Furthermore, the activity level of the iModulon was high in AlgU expression strains in comparison to all other samples in the data set (Fig. 1f).

Unlike the traditional differential expression of genes (DEG) analysis which only compares two conditions and therefore captured signals from several related transcriptional mechanisms, our approach used all available data to separate each signal into its own iModulon. This global use of available data allows us to provide and strengthen evidence of gene-regulator interactions and more easily quantify the global transcriptional state of the cell. The key iModulons that are used to interpret the transcriptomic changes are summarized in the Data S1.

## iModulon activities reveal transcriptional changes in *P. syringae* during coculture with other species

Interspecies interactions in cocultures result in altered transcriptome composition. The production of exometabolites in community settings has been shown to be a competitive strategy employed for exploitation and interference in the presence of neighbors (10, 19). It has been found that bacteria are primed for competition via exometabolite

production in growth-limiting conditions such as in natural environments that harbor microbial communities (10, 11). A previous study investigated the effects of exometabolite interactions during stationary phase between three bacterial strains found in terrestrial environments: *P. syringae*, *B. thailandensis*, and *C. violaceum* (10). This study revealed that coculture with *B. thailandensis* elicits strong transcriptional responses in both *P. syringae* and *C. violaceum* (10). Analysis of iModulon activities allows us to untangle the composition of the transcriptome and more easily interpret the behavior of bacteria in complex microbial communities, thereby providing insight into various exploitation and survival strategies.

Over a period of 45 h (exponential phase: 0–12.5 h, stationary phase: 12.5–45 h), iModulon activity levels were notably altered in cocultures containing *B. thailandensis*, whereas *C. violaceum* elicited a weaker transcriptional response (Fig. 2a). *B. thailandensis* is known to produce several bioactive exometabolites such as antibiotics that are harmful to its neighbors (10, 11). As a result, *P. syringae* upregulated the Resistance-1 iModulon in the presence of *B. thailandensis*, which contains genes encoding resistance-nodulation-division (RND) and Major Facilitator Superfamily efflux pumps that confer antibiotic resistance.

The activity of a second resistance-associated iModulon, Resistance-2, was found to have a strong positive correlation with that of the Resistance-1 iModulon in the monoculture and coculture samples (Fig. 2d). Notably, this iModulon was upregulated in the presence of *B. thailandensis* as well as *C. violaceum* (Supplementary Note S1). We believe that while the upregulation of the Resistance-1 iModulon is necessary in the presence of more harmful compounds such as antibiotics produced by *B. thailandensis*, the Resistance-2 iModulon is initially upregulated to confer antibiotic resistance up to a certain extent. The Resistance-2 iModulon contains genes involved in the oxidative stress response, copper efflux, and a gene encoding an RND efflux pump neighboring the copper efflux genes.

Previous studies have identified a co-selection of antibiotic and metal resistance in other organisms, with the co-selection of the two increasing the ability of bacteria to resist antibiotics (20, 21). Additionally, this co-selection has been found to be stronger under the pressure of oxidative stress. The upregulation of the AlgU iModulon, which includes genes involved in oxidative stress resistance, in the presence of *B. thailandensis* indicates that *B. thailandensis* creates oxidative stress (Fig. 2e). Altogether, the observed iModulon membership and activities add to existing knowledge regarding the co-selection of metal and antibiotic resistance in environments of oxidative stress. The regulator(s) underlying the Resistance iModulons should be identified and characterized in future studies.

Interestingly, we found a strong negative correlation between the activities of the iModulons involved in flagellar assembly, namely Flagella-1 and Flagella-2, and that of the Resistance-1 iModulon in the monoculture and coculture samples (Fig. 2b and c). As proton motive force powers both efflux pumps as well as flagellar assembly, we hypothesize that there exists a transcriptomic trade-off between motility and the powering of efflux pumps in conditions of antibiotic stress (22, 23). Alternatively, the reduction in the motility of *P. syringae* may be a result of the competitive strategies of *B. thailandensis*. The production of compounds that inhibit motility in neighbors has been observed in *Burkholderia pseudomallei*, a close relative of *B. thailandensis* (24). As swimming motility confers partial resistance to antibiotics, the downregulation of flagellar genes may be a strategy of interference employed by *B. thailandensis* to increase the susceptibility of *P. syringae* to antibiotics (25).

Cocultures containing *B. thailandensis* were also found to have low Pyoverdine and Yersiniabactin iModulon activities (Fig. 2f and g). Through a coexpression network analysis, a previous study observed that the downregulation of the pyoverdine receptor, PSPTO_1206, was coordinated with the upregulation of its closest homolog in *B. thailandensis* (10). These previous findings suggest that *B. thailandensis* may attempt to utilize siderophores produced by *P. syringae*. *P. aeruginosa* has been found to adapt to the

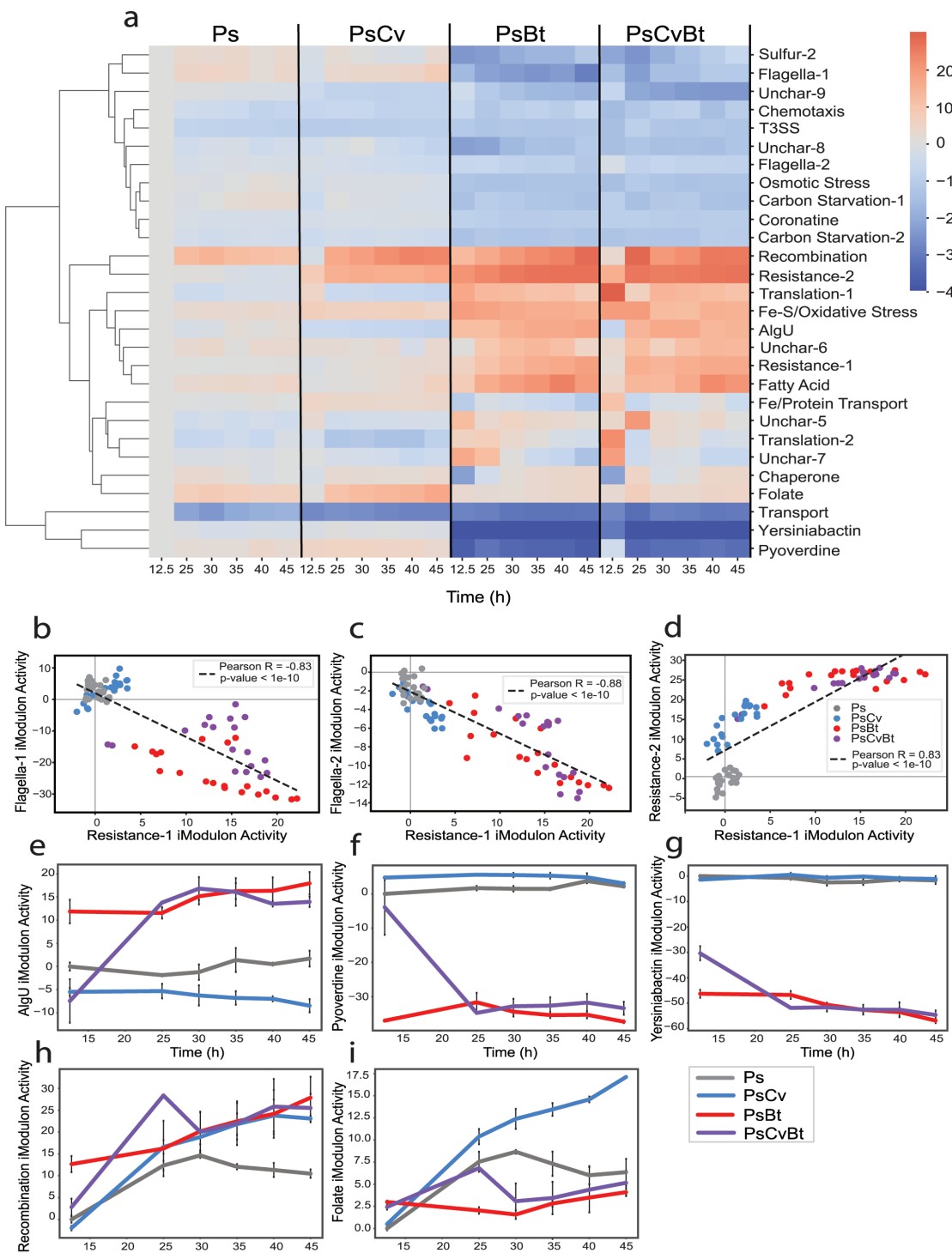

**FIG 2** iModulons characterize the transcriptional response of *P. syringae* in bacterial community settings (Ps: *P. syringae* monoculture, PsCv: *P. syringae-C. violaceum* coculture, PsBt: *P. syringae-B. thailandensis* coculture, PsCvBt: *P. syringae-C. violaceum-B. thailandensis* coculture). (a) Clustermap of changes in the activity of selected iModulons over a period of 45 h (exponential phase: 0–12.5 h, stationary phase: 12.5–45 h) based on community membership. (b–d) Scatter plots of the activities of two iModulons: Resistance-1 and Flagella-1 iModulons (b), Resistance-1 and Flagella-2 iModulons (c), and Resistance-1 and Resistance-2 iModulons (d) for monoculture and coculture conditions. (e–i) Time courses of the activity of the AlgU, Pyoverdine, Yersiniabactin, Recombination, and Folate iModulons in different community settings (*n* values are listed in Table S2).

presence of siderophore cheaters by reducing pyoverdine production and increasing other methods of iron acquisition (26). Based on the downregulated activity levels of the Pyoverdine iModulon and the relatively higher activity levels of the Fe/Protein Transport iModulon (Fig. 2a), *P. syringae* appears to downregulate biosynthesis, transport, and uptake of pyoverdine through the downregulation of the Pyoverdine iModulon in order to deprive *B. thailandensis* of siderophores. Additionally, based on the similar trend of downregulated activity levels of the Yersiniabactin iModulon, consisting of genes involved in the biosynthesis and uptake of the siderophore yersiniabactin, *P. syringae* may be employing a similar strategy in depriving *B. thailandensis* of siderophores it may attempt to utilize. Alternatively, the downregulation of both siderophore-associated iModulons may indicate that *P. syringae* experiences iron sufficiency in the presence of *B. thailandensis*. This would result in Fur-mediated downregulation of siderophore biosynthesis (27). It is notable that the Fe-S/Oxidative Stress iModulon is upregulated in cocultures containing *B. thailandensis*, as an abundance of iron can contribute to the production of reactive oxygen species (ROS) through the Fenton reaction.

The activity of the Recombination and Folate iModulons display upregulated transcriptional responses of *P. syringae* common to the monoculture and coculture conditions, albeit to varying degrees (Fig. 2h and i). The Recombination iModulon consists of genes encoding recombinase family proteins along with several genes of unknown functions. Hence, this iModulon may capture the response of the bacteria to DNA damage, either due to competitive neighbors or due to cell death commonly observed during stationary phase (28). The upregulation of the Recombination iModulon may also be due to oxidative stress, as evidenced by the upregulation of the Fe-S/Oxidative Stress iModulon in coculture conditions. The similar activity of the iModulon in the coculture conditions indicates that DNA recombination in coculture does not depend on the species-specific competition strategies of these organisms. The Folate iModulon, containing genes involved in the production of tetrahydrofolate, was also upregulated in all four conditions. However, the activity of the iModulon in the presence of *B. thailandensis* was lower in comparison to the activity of the iModulon in its absence. Since tetrahydrofolate plays an important role in purine and pyrimidine synthesis, the higher activity of the iModulon in the absence of *B. thailandensis* suggests that nucleotide synthesis is upregulated to a lower extent as a result of the competition strategies of *B. thailandensis*. (29). iModulons can thus describe in detail how the transcriptome composition changes in microbial communities.

## Alterations in iModulon activities reveal transcriptional responses of *P. syringae* to the plant immune system

The first line of defense against pathogens in plants involves the recognition of pathogen-associated molecular patterns (PAMPs) by pattern recognition receptors (PRRs) (30, 31). Upon recognition of PAMPs by PRRs, pattern triggered immunity (PTI) is initiated in plant cells, resulting in various processes including ROS burst and upregulated transcription of defense-associated genes (31). The conserved flagellin-derived peptide flg22 found in the filament of the bacterial flagellum is a PAMP recognized by the PRR FLS2 (32). In order to suppress PTI and aid in infection, pathogens secrete effectors through the T3SS (30). However, plants have evolved to detect effectors through intracellular nucleotide-binding domain leucine-rich repeat containing receptors, which subsequently initiates effector triggered immunity (ETI) (31). Both PTI and ETI are thought to have common downstream components, with ETI upregulating parts of the PTI response, allowing the plant to effectively resist the pathogen. Although the separation of the two immune responses is difficult, previous studies have investigated PTI and ETI responses through the infiltration of PTI elicitors (e.g., flg22 and chitosan) and ectopic expression of effectors (e.g., AvrRpt2), respectively, in addition to various immune system-associated plant mutations (12, 13). The analysis of iModulons adds to the existing knowledge of plant-pathogen interactions at a systems level.

Compared to naive host infection, exposure to PTI through pretreatment with flg22 caused a downregulation of pathogenesis-associated iModulons (T3SS and Coronatine) and translation-associated iModulons (Translation-1 and Translation-2) (Fig. 3a). In contrast, stress-related iModulons (Resistance-1, Nitrogen, AlgU, and Osmotic Stress) were upregulated. This observation suggests that priming the plant's PTI induced higher stress in *P. syringae*. In agreement with previous studies, we found that exposure to PTI also upregulated motility, as seen through the upregulation of the Chemotaxis, Flagella-1, and Flagella-2 iModulons (33). It is not clear as to whether this upregulation of motility is due to misregulation in transcription caused by PTI or evolved mechanisms attempting to move to a more favorable environment.

Previous studies have found that pre-induction of PTI results in reduced induction of T3SS-associated genes compared to naive host infection (13). A mechanism by which *A. thaliana* suppresses the T3SS of *P. syringae* is through the production of the T3SS-suppressing electrophilic compound sulforaphane (34, 35). However, *P. syringae* possesses genes involved in the detoxification of this compound, which are present in the Resistance-1 iModulon (PSPTO_3100, *saxA*; PSPTO_1858, *saxF*). As these genes allow the bacteria to resist the effects of sulforaphane, we analyzed the activities of the T3SS and Resistance-1 iModulons over a period of 5 h post inoculation (hpi) during naive host infection and pre-induced PTI (Fig. 3b). In naive host infection, *P. syringae* mounts its attack by rapidly upregulating the T3SS iModulon up to 3 hpi. Between 3 hpi and 5 hpi, both the T3SS and Resistance-1 iModulons are downregulated. This response may be due to the establishment of a colony, allowing cells to shift their transcriptional allocation away from virulence mechanisms. However, in the pre-induced PTI condition, *P. syringae* instead upregulates the Resistance-1 iModulon more strongly, in a more defensive strategy that likely arises due to the increased infection-readiness of the plant. By 5 hpi, it appears that *saxAF* has successfully detoxified some sulforaphane, as evidenced by the downregulation of the Resistance-1 iModulon at this time point. Alternatively, the downregulation of the T3SS iModulon by 5 hpi may indicate that *P. syringae* is unable to overcome the effects of sulforaphane.

Salicylic acid (SA)-mediated signaling pathways contribute to the PTI response in plants, and its accumulation has been found to enhance the sensitivity of plants to PAMPs (36, 37). Hence, we compared the response of *P. syringae* to infection in *A. thaliana* pre-treated with the plant defense hormone, SA, and the PAMP, chitin, and found that both conditions upregulated starvation responses (Carbon Starvation-1, Carbon Starvation-2, and Nitrogen iModulons) and motility (Flagella-1, Flagella-2, ad Chemotaxis iModulons) in *P. syringae* (Fig. 3c) (12, 37). While the Carbon Starvation-1 iModulon contains the gene encoding carbon starvation protein A (PSPTO_4638) and others involved in acetate utilization, the Carbon Starvation-2 iModulon contains genes involved in the utilization of alternative carbon sources such as inositol, arabinose, and possibly pectin (Supplementary Note S2). Additionally, the Nitrogen iModulon contains genes involved in the nitrogen stress response and utilization of alternate nitrogen sources such as urea, pyrimidines, nucleosides, and amino acids (Fig. S1). Previous studies have identified mechanisms of PTI-induced pathogen starvation in *A. thaliana*, such as the removal of sugars from the pathogen-colonized apoplast (38). As SA is known to induce PTI regulators, the upregulation of these three iModulons during both SA treatment and chitin-induced PTI may indicate that SA plays a role in depriving pathogens of carbon and nitrogen sources during PTI.

Similar to PTI, ETI induction through the ectopic expression of the effector AvrRpt2 in *P. syringae* elicited a strong response in the bacteria (Fig. 3d). The reduced iron acquisition ability during AvrRpt2-induced ETI and pre-induced PTI using chitin was observed through the consistent downregulation of the Pyoverdine and Fe/Protein Transport iModulons. Additionally, the AlgU and Osmotic Stress iModulons were largely upregulated during AvrRpt2-induced ETI and chitin-induced PTI. However, samples obtained through infection in an *A. thaliana* mutant lacking the SA receptor *npr1* had differentially

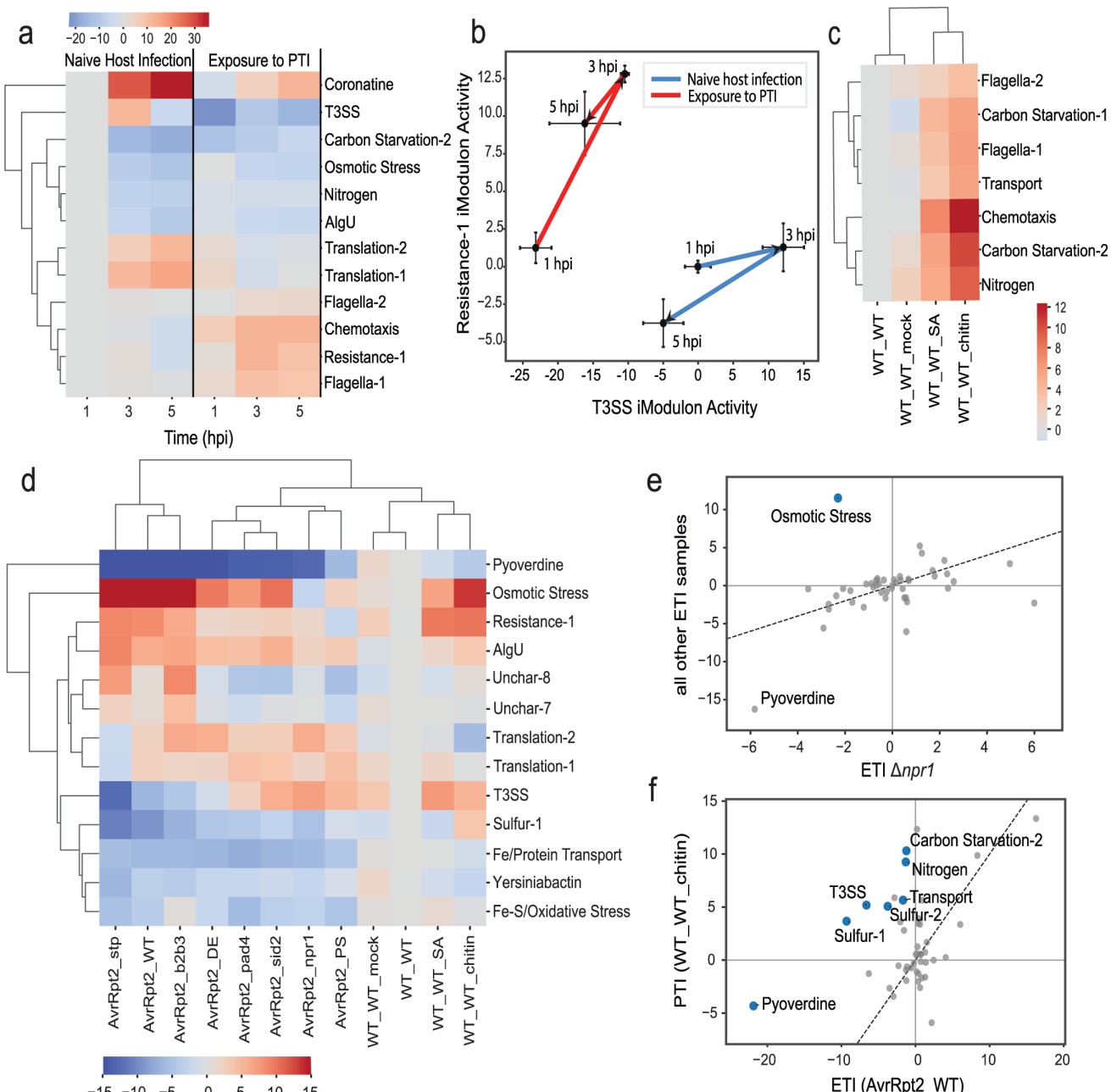

**FIG 3** Changes in iModulon activity levels advance our understanding of plant-pathogen interactions (hpi: hours post inoculation, PTI: pattern triggered immunity, ETI: effector triggered immunity). Full sample descriptions are listed in Table S3. (a) Clustermap of changes in the activity of selected iModulons over a period of 5 hpi during naive host infection and exposure to PTI through pretreatment with flg22. (b) Dynamic changes in the activity of the T3SS and Resistance-1 iModulons during naive host infection and exposure to PTI through pretreatment with flg22 (*n* = 3). (c) Clustermap of changes in activities of selected iModulons during infection in *A. thaliana* pretreated with different infiltration agents (WT_WT: none, WT_WT_mock: water, WT_WT_SA: salicylic acid, WT_WT_chitin: chitin). (d) Clustermap of changes in activity of selected iModulons during AvrRpt2-elicited ETI in different *A. thaliana* mutants and during infection in A. thaliana pretreated with different infiltration agents. (e) Differentially activated iModulons in AvrRpt2-elicited ETI samples in npr1 *A. thaliana* mutants compared to all other AvrRpt2-elicited ETI samples (threshold = 5 and fdr = 0.35). (f) Differentially activated iModulons during exposure to PTI using chitin compared to AvrRpt2-elicited ETI (threshold = 5 and fdr = 0.35).

downregulated Osmotic Stress iModulon activity in comparison to all other AvrRpt2-induced ETI samples, where the iModulon was upregulated (Fig. 3e). Hence, we hypothesize that the SA receptor npr1 is necessary for the plant to create an environment of osmotic stress.

Although there is an overlap in the outputs of PTI and ETI, several iModulons were differentially activated while comparing samples obtained during pre-induced PTI using chitin and AvrRpt2-induced ETI (Fig. 3f). Sulfur, carbon, and nitrogen starvation may be features specific to pre-induced PTI, as seen through the differential upregulation of the Sulfur-1, Sulfur-2, Carbon Starvation-2, and Nitrogen iModulons in pre-induced PTI samples in comparison to AvrRpt2-induced ETI samples. Additionally, there was a stronger downregulation in the Pyoverdine iModulon during AvrRpt2-induced ETI compared to pre-induced PTI. Furthermore, the T3SS iModulon was differentially downregulated during AvrRpt2-induced ETI compared to pre-induced PTI. Thus, dimensionality reduction via iModulon analysis effectively summarized the transcriptomic changes underlying the complex interactions between this pathogen and multiple plant immunity systems.

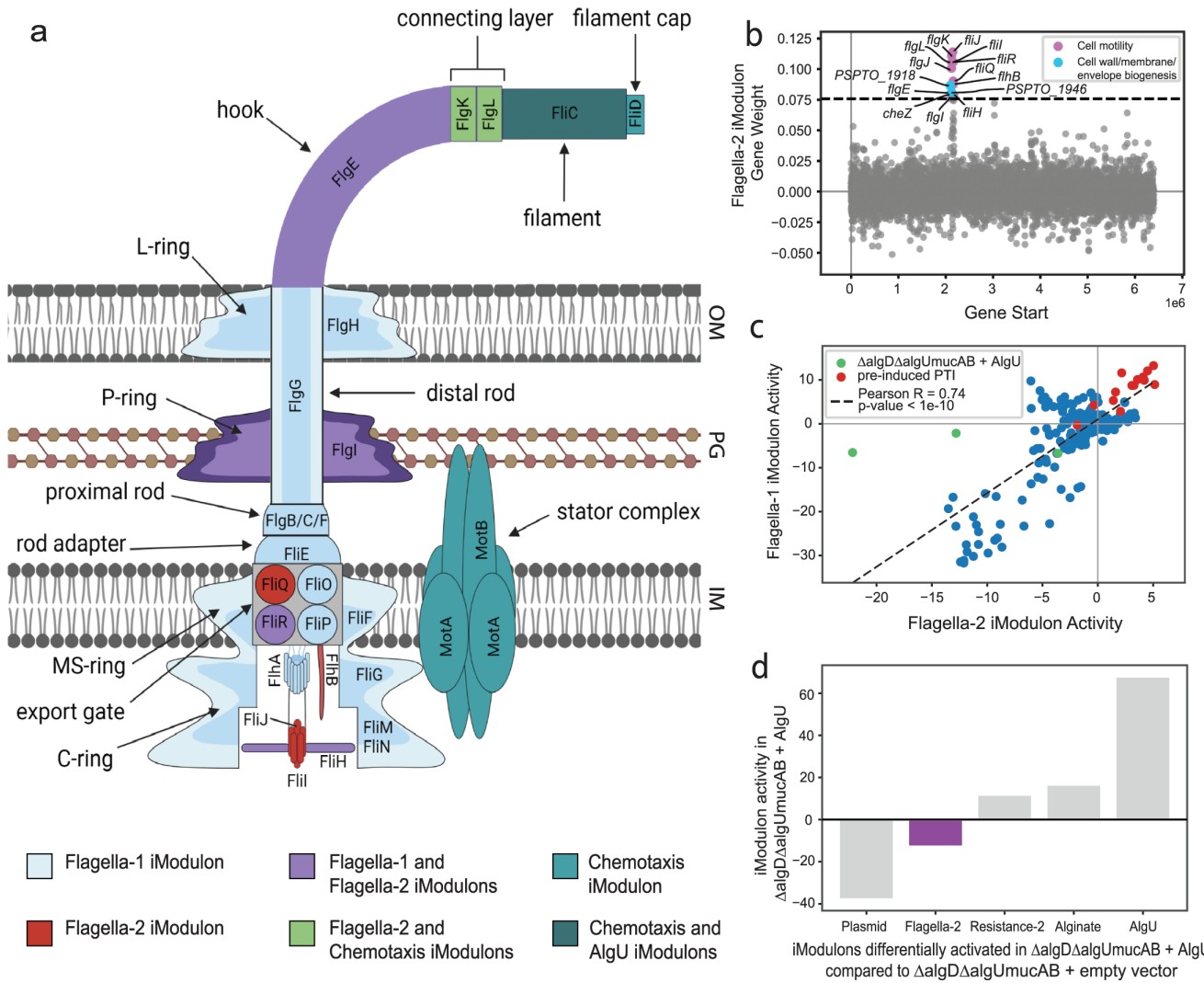

**FIG 4** iModulons reveal additional details in the AlgU-dependent downregulation of flagellar genes. (a) Schematic representation of the general structure of the bacterial flagellum with genes categorized based on iModulon membership (OM: outer membrane, PG: peptidoglycan layer, IM: inner membrane) (41). This figure was created using BioRender (BioRender.com). (b) Flagella-2 iModulon gene membership. (c) Scatter plot of Flagella-1 and Flagella-2 iModulon activities. (d) Differentially activated iModulons in ΔalgDΔalgUmucAB mutants transformed with an AlgU expression vector in comparison to mutants transformed with an empty vector (reference condition; threshold = 5 and fdr = 0.15).

## iModulons show fine tuning of AlgU-dependent downregulation of flagellar genes *in planta*

In order to evade PTI, *P. syringae* attempts to reduce the expression of flagella in the apoplast by downregulating flagellar genes in an AlgU-dependent manner (39). While some studies have found that *fliC* in particular is targeted by AlgU, others have determined multiple flagellar genes under the regulation of AlgU (33, 40). Our analysis of flagellar assembly associated iModulons provide additional details on possible AlgU-dependent regulatory mechanisms of flagellar gene expression.

The structural components of the flagellum are largely encoded by the genes in the Flagella-1 iModulon (Fig. 4a). However, a few key genes encoding proteins involved in flagellar assembly belong to the Flagella-2 iModulon (Fig. 4b). Besides encoding structural parts of the flagella in the extracellular region, the iModulon contains genes encoding proteins that control protein translocation through the central channel of the flagellum, thereby controlling the construction of distal parts of the flagellum. The downregulation of the Flagella-2 iModulon would likely prevent the formation of the extracellular part of the flagellum that can be detected by PRRs, allowing bacteria to evade detection by the plant.

Although the two flagella iModulons were highly correlated in almost all samples (Fig. 4c), they were differentially activated in *algU mucAB algD* mutants transformed with an *algU* expression vector *in planta* in comparison to mutants with an empty vector; this condition downregulated Flagella-2 only (Fig. 4d). Considering the previously identified role of AlgU in the evasion of PTI by *P. syringae in planta*, these results suggest that AlgU exerts a strong negative regulatory effect on the Flagella-2 iModulon genes, which would prevent the production of extracellular parts of the flagella that would trigger PTI. Additionally, in agreement with previous studies showing that pre-induced PTI causes misregulation of flagellar genes by AlgU, we found that the AlgU-dependent downregulation of the Flagella-2 iModulon was absent in samples obtained during pre-induced PTI using flg22 and chitosan. Thus, iModulons provide nuance to the regulation of flagellar component expression, and motivate more detailed studies to determine the interaction of regulators at flagellar promoters.

## DISCUSSION

In the present study, we revealed the TRN of *P. syringae* using machine learning on all available data. We identified 45 iModulons, which together explain 64% of the variance in gene expression. Twenty-eight of the iModulons represent sets of genes involved in particular biological functions. iModulons thus represent biologically interpretable responses of the TRN. Several of these iModulons were found to have notably altered activity levels in two- or three-member cocultures, revealing strategies of competition and survival involved in interspecies interactions. We also advanced the current understanding of plant-pathogen interactions by uncovering the transcriptional responses of *P. syringae* to the plant immune system. Finally, we identified an AlgU-dependent differential regulation of flagellar genes, allowing *P. syringae* to evade the plant immune system. Through iModulon analysis, the findings presented in this study notably deepen existing knowledge and provide new hypotheses that can be addressed in future studies. Researchers interested in particular genes or regulons can browse or search all of the iModulon data for this study, which is available on iModulonDB.org.

As an advancement over previous iModulon studies, the present study analyzes transcriptional responses during interactions with other organisms, both in microbial communities and in plant infections. We first implemented iModulon analysis to explore bacterial interactions in cocultures, and uncovered possible mechanisms of competition and survival in *P. syringae* and *B. thailandensis,* thereby demonstrating a novel method of studying complex interactions in bacterial communities. We identified a fight-or-flight response involved in the ability of *P. syringae* to survive in environments of antibiotic stress, captured through the activities of the Flagella-1, Flagella-2, and

Resistance-1 iModulons. Additionally, the activity of a second resistance-associated iModulon, Resistance-2, points to the involvement of copper transport-related genes in the antibiotic resistance response of *P. syringae*. Furthermore, we identified a possible mechanism of adaptation to the presence of siderophore cheaters at the transcriptional level through the downregulation of the Pyoverdine and Yersiniabactin iModulons.

The analysis of iModulon activity levels also indicated a strong transcriptional response from *P. syringae* to the *A. thaliana* immune system. In addition, we identified a component of the SA signaling pathway in *A. thaliana*, namely npr1, that may contribute to inducing osmotic stress in the pathogen-colonized apoplast (42). We also discovered differences between the outputs of the PTI and ETI responses in *A. thaliana*, such as the induction of starvation in pathogens during PTI through the differential activation of iModulons associated with carbon, nitrogen, and sulfur starvation responses. Finally, our study highlights the utility of iModulon analysis for identifying nuanced regulatory mechanisms through the analysis of flagellar gene regulation mediated by AlgU as a means of evading the plant immune system. Altogether, our findings offer good prospects for the use of iModulons in teasing out complex interactions relevant to other microbial communities and infections, which can be improved upon by analyzing each member of the community as opposed to focusing on just one.

The ICA algorithm used to identify iModulons assumes that the activity of each regulator is statistically independent from one another, has a non-Gaussian distribution, and affects genes in a linear manner (3). While these assumptions allow ICA to outcompete other algorithms in quantitatively representing the TRN, more complex regulatory mechanisms may not be captured (43). Additionally, our decomposition of the *P. syringae* TRN does not explain 36% of the variance in the data set. This is due to a lack of clear co-regulation, and is generally assumed to be noise. Future research can work towards a more comprehensive decomposition of the *P. syringae* TRN through the inclusion of more data (5, 44). Appending additional data generated under diverse conditions to the current data set may result in a change in the set of iModulons identified by ICA, such as new iModulons being identified, large multi-purpose iModulons splitting into iModulons representing each functionality individually, and reduction of noise.

In conclusion, we have demonstrated the utility of iModulon analysis to achieve a deep understanding of the complex interactions that take place between pathogens and their hosts, as well as in microbial communities. The elucidation of the TRN structure of *P. syringae* using iModulons also facilitates future studies focused on TF identification and characterization of unknown genes. Together, this study provides the foundation for a quantitative TRN of *P. syringae*.

## MATERIALS AND METHODS

The methods followed in this study for RNA-seq data acquisition and processing, performing ICA, and iModulon characterization were adapted from Sastry et al. (17).

### RNA-seq data acquisition, processing, and quality control

The PyModulon workflow was first used to compile the metadata for all publicly available data in the NCBI SRA database (https://www.ncbi.nlm.nih.gov/sra) for *P. syringae* pv *tomato* DC3000 as of August 20, 2020. A total of 202 samples were collected from previous studies (10–14, 16). Further, the Nextflow pipeline was run using Amazon Web Services to process the data set as previously described (17, 45). Raw FASTQ files available on NCBI were downloaded using fasterq-dump (https://github.com/ncbi/sra-tools/wiki/HowTo:-fasterq-dump), after which Trim Galore (https://www.bioinformatics.babraham.ac.uk/projects/trim_galore) was used for read trimming. The trimmed reads were checked using FastQC (http://www.bioinformatics.babraham.ac.uk/projects/fastqc/), and then aligned to the reference genome using Bowtie (46). RSEQC was performed to determine the direction of the reads, after which featureCounts was used to generate read counts (47, 48). MultiQC was used to ensure the quality of the data set,

and the data were subsequently converted into units of $\log_2$ transcripts per million (TPM) (49).

Hierarchical clustering was performed to visualize global correlations between each sample with all other samples, after which samples with atypical expression profiles were discarded. Further, sample metadata were manually curated, and biological replicates with low correlation were dropped ($R^2$ <0.95 for non-coculture samples, $R^2$ <0.90 for coculture samples). Additionally, samples with no replicates were discarded. Finally, log-TPM data within projects were normalized to reference conditions specific to each project.

## Independent component analysis (ICA)

ICA was performed to decompose the matrix of gene expression profiles (X) into a matrix of independent components (i.e., iModulons) (M) and their activities in all conditions (A) (17). Specifically, the Scikit-learn algorithm, FastICA, was performed 100 times with a convergence tolerance of $10^{-7}$ (50). The number of components was set based on the number of components required to reconstruct 99% of variance in the data through PCA. The independent components (ICs) thus obtained were clustered using DBSCAN to determine robust ICs, with an epsilon of 0.1 and minimum cluster seed size of 50 (51). To ensure the reproducibility of the clusters, the signs of gene weights were inverted such that genes with the highest weight in each cluster had a positive sign. The centroids of these clusters were taken as the final ICs. Additionally, the optimal dimensionality was determined to be 180 by clustering the gene expression profile several times between the dimensions of 10 and 200 (52). The threshold to determine gene members for the 45 iModulons obtained was determined based on a D'Agostino kurtosis test.

## Gene annotation compilation and regulator enrichment

The pipeline for gene annotation has been detailed at https://github.com/SBRG/pymodulon/blob/master/docs/tutorials/creating_the_gene_table.ipynb. Gene annotations were primarily obtained from AE016853.1, with additional information being collected from KEGG (and Cluster of Orthologous Groups (COG) using EggNOG mapper (53). Additionally, the Uniprot ID mapper was used to obtain the Uniprot IDs (54). Furthermore, the TRN was annotated using RegPrecise, Pseudomonas Genome Database, and literature (55, 56).

For each iModulon, Fisher's exact test with Benjamini-Hochberg false discovery rate (FDR) correction was used to calculate regulator enrichment against known regulons (FDR = $10^{-5}$) as well as KEGG and GO enrichment (FDR < $10^{-2}$). iModulons were subsequently named based on their enriched regulators or functional characteristics.

## Differential iModulon activation

Differences in iModulon activation between two or more conditions were calculated by using a log-normal distribution. To determine the statistical significance of these differences, $P$-values were determined by calculating the absolute value of the difference between mean iModulon activities and comparing with the log-normal distribution of the iModulon. This was followed by Benjamini-Hochberg correction. A threshold of 5 was considered to be significant.

## Generating iModulon activity clustermaps

iModulon activity levels were clustered hierarchically using the average linkage method. Additionally, the minimum activity level for Fig. 2a was set to −40, whereas an activity range of −15 to 15 was selected while creating Fig. 3d. The activity levels of iModulons for these clustermaps have been listed in Data S2 and Data S3, respectively.

## Generating iModulon dashboards

We generated the iModulonDB dashboards using the PyModulon package (9, 17). The pipeline for the same can be found at https://pymodulon.readthedocs.io/en/latest/tutorials/creating_an_imodulondb_dashboard.html.

### ACKNOWLEDGMENTS

This research received no specific grant from any funding agency in the public, commercial, or not-for-profit sectors.

A.V.S. and B.O.P. conceptualized the work; C.R.L. curated the data; H.B. analyzed the data and drafted the paper; K.R. and B.O.P. provided mentorship and guidance throughout; K.R. and H.B. built the webpage. All participated in reviewing and editing the paper.

### AUTHOR AFFILIATIONS

[1]Department of Bioengineering, University of California San Diego, La Jolla, California, USA

[2]Department of Pediatrics, University of California San Diego, La Jolla, California, USA

[3]Bioinformatics and Systems Biology Program, University of California San Diego, La Jolla, California, USA

[4]Center for Microbiome Innovation, University of California San Diego, La Jolla, California, USA

[5]Novo Nordisk Foundation Center for Biosustainability, Kongens Lyngby, Denmark

### PRESENT ADDRESS

Kevin Rychel, Palmona Pathogenomics, Inc, Menlo Park, California, USA
Anand V. Sastry, Absci, Vancouver, Washington, USA

### DATA AVAILABILITY

The data used for this study are available at the accession numbers listed in the Supplementary Data. Additionally, various files including the X, M, and A matrices, along with the source code for iModulon analysis and creating figures, are available on GitHub. Code for the analysis pipeline can be found on GitHub.

### ADDITIONAL FILES

The following material is available online.

#### Supplemental Material

**Data S1 (mSystems00437-23-s0001.xlsx).** Sheet 1: RNA-seq sample list. Sheet 2: description of key iModulons.
**Data S2 (mSystems00437-23-s0002.csv).** Activity levels used for Figure 2a.
**Data S3 (mSystems00437-23-s0003.csv).** Activity levels used for Figure 3d.
**Figure S1 (mSystems00437-23-s0004.eps).** Genes in the Nitrogen iModulon represent various nitrogen sources used under nitrogen starvation.
**Supplemental Material (mSystems00437-23-s0005.docx).** Supplemental notes, figure caption, tables, and references.

#### Open Peer Review

**PEER REVIEW HISTORY (review-history.pdf).** An accounting of the reviewer comments and feedback.

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
