## [Reviewer comments · mSystems]

Machine learning uncovers the *Pseudomonas syringae* transcriptome in microbial communities and during infection

Heera Bajpe, Kevin Rychel, Cameron Lamoureux, Anand Sastry, and Bernhard Palsson

Corresponding Author(s): Bernhard Palsson, UCSD

Review Timeline:

Submission Date:	May 9, 2023
Editorial Decision:	June 8, 2023
Revision Received:	June 28, 2023
Accepted:	July 19, 2023

Editor: Ryan McClure

Reviewer(s): The reviewers have opted to remain anonymous.

Transaction Report:

DOI: <https://doi.org/10.1128/msystems.00437-23>

June 8, 2023

Prof. Bernhard O Palsson
UCSD
San Diego

Re: mSystems00437-23 (Machine learning uncovers the *Pseudomonas syringae* transcriptome in microbial communities and during infection)

Dear Prof. Bernhard O Palsson:

Thank you for submitting your manuscript to mSystems. We have completed our review and I am pleased to inform you that, in principle, we expect to accept it for publication in mSystems. However, acceptance will not be final until you have adequately addressed the reviewer comments.

Preparing Revision Guidelines

Please return the manuscript within 60 days; if you cannot complete the modification within this time period, please contact me. If you do not wish to modify the manuscript and prefer to submit it to another journal, please notify me of your decision immediately so that the manuscript may be formally withdrawn from consideration by mSystems.

Sincerely,

Ryan McClure

Editor, mSystems

Journals Department
Reviewer comments:

Reviewer #1 (Comments for the Author):

Comments to the authors:

In this well-written manuscript, the authors describe how they analyzed the transcriptional regulatory network of *Pseudomonas syringae* pv. tomato DC3000 using publicly available RNA-Seq data to identify 44 iModulons. Each iModulon represents a group of genes whose expression appears to be coordinately regulated across diverse physiological conditions. The authors speculate on the observed patterns of gene regulation in relation to growth of *P. syringae* in the presence of one or two other bacteria, in response of *Arabidopsis thaliana* host defenses (using some mutant strains) and in relation to AlgU control of flagellar genes. Suggestions are offered for the authors' consideration.

1. p. 3, lines 3-7: Here the authors briefly described the experimental context of the data under study (plant-immunity inducing conditions, bacterial mutations and coculturing). While quite a bit of information is available in the supplemental materials (Tables S1-S3), it might help to have some sort of an abbreviated table in the main text summarizing the key parameters, including in vivo or in vitro growth conditions, that were experimentally varied to help frame the experimental data used to identify the iModulons for the reader upfront.

2. p. 4, last paragraph: The authors discuss the downregulation of the flagella iModulons in the context of efflux pumps and antibiotic resistance. There is no discussion in the manuscript about biofilm formation in *P. syringae*. Motility and biofilm formation are often inversely regulated. Were the biofilm genes not within an iModulon and therefore not part of the analysis? This raises the question, are there key physiological outputs that were not captured within iModulons (i.e., what is missed)? What about the 36% of gene expression not included (p. 7 second to last line). Some discussion of the limitations of the iModulon approach seems important to include.

3. p. 5, end of second full paragraph: The text states that "production of tetrahydrofolate was upregulated in all four conditions" and that "tetrahydrofolate plays an important role in purine and pyrimidine synthesis". However, the text then goes on to say that "nucleotide synthesis DECREASES". These statements seem contradictory and it is recommended that the text be rewritten here.

4. The references need to be further proofed for consistent use of italics for organism names and correct use of capital letters in titles (only the first word).

Reviewer #2 (Comments for the Author):

See attached PDF file.

Comments on the manuscript “Machine learning uncovers the *Pseudomonas syringae* transcriptome in microbial communities and during infection”

In the manuscript “Machine learning uncovers the *Pseudomonas syringae* transcriptome in microbial communities and during infection”, Bajpe and colleagues use publicly available RNA seq datasets to describe the transcriptome of *P. syringae* upon interaction with other bacterial species and during infection of *Arabidopsis thaliana*, and map transcriptional networks.

This study provides the first thorough analysis of gene regulation occurring when *P. syringae* interacts with other microbes and infects plants, thereby advancing the basic knowledge of how the bacteria reprograms its gene expression in response to environmental cues typically encountered during its lifestyle, and how this benefits its interaction with other bacteria and plants. Such analysis will provide key resources for researchers willing to investigate the molecular basis of some of the transcriptional regulations described in the paper.

The manuscript is very well written and scientifically sound. I have very few comments, most of them related to the resolution of the figures, which should be increased, and to data interpretation.

Comments

- 1) The resolution of the figures should be increased as most of them appear pixelated on the printed version of the manuscript or upon zooming.
- 2) **Supplementary Note S1.** The authors claim that *C. violaceum* was found to upregulate the activities of the Resistance-1 and Resistance-2 iModulons in *P. syringae*. However, according to the heatmap in Figure 2a, only Resistance-2 iModulon, and not Resistance-1, is upregulated when *P. syringae* interacts with *C. violaceum*. The authors should change or further clarify their data interpretation in the text.
- 3) **Results. Lines 174-186.** While the decrease of expression of siderophore genes in the presence of *B. thailandensis* may reflect a strategy adopted by *P. syringae* for decreasing siderophore production to prevent piracy, as suggested by the authors, it could also indicate that *P. syringae* somehow experiences iron sufficiency when cocultured with *B. thailandensis*, therefore promoting Fur-mediated transcriptional repression of siderophore genes. Iron sufficiency, and especially overabundance, can also promote the formation of free radicals through the Fenton reaction, which could also explain the induction of the Fe-S/Oxidative Stress iModulon in *P. syringae* upon coculture with *B. thailandensis*. The authors should maybe consider evoking these possibilities in the text.
- 4) **Results. Lines 188-199.** The authors suggest that the induction of the Recombination iModulon may capture the response of the bacteria to DNA damages. One may consider the possibility of these DNA damages to be the

result of oxidative stress experienced by *P. syringae* when cocultured with *B. thailandensis*, as suggested by the induction of the Fe-S/Oxidative Stress regulon. The authors should discuss this possibility in the text.

Minor comments

- **Results. Line 142.** “[...] **RND** and **MFS** efflux pumps [...]”. Authors should spell out the RND (Resistance-Nodulation- Division) and MFS (Major Facilitator Superfamily) abbreviations the first time they appear in the text.
- **Figure 2a.** The Ps, PsCv, PsBt, and PsCvBt abbreviations should be spelled out in the figure legend.
- **Supplementary Note S1.** “*C. violacium*”. Replace “*violacium*” with “*violaceum*”.

Reviewer #1 (Comments for the Author):

In this well-written manuscript, the authors describe how they analyzed the transcriptional regulatory network of *Pseudomonas syringae* pv. tomato DC3000 using publicly available RNA-Seq data to identify 44 iModulons. Each iModulon represents a group of genes whose expression appears to be coordinately regulated across diverse physiological conditions. The authors speculate on the observed patterns of gene regulation in relation to growth of *P. syringae* in the presence of one or two other bacteria, in response of *Arabidopsis thaliana* host defenses (using some mutant strains) and in relation to AlgU control of flagellar genes. Suggestions are offered for the authors' consideration.

1. p. 3, lines 3-7: Here the authors briefly described the experimental context of the data under study (plant-immunity inducing conditions, bacterial mutations and coculturing). While quite a bit of information is available in the supplemental materials (Tables S1-S3), it might help to have some sort of an abbreviated table in the main text summarizing the key parameters, including in vivo or in vitro growth conditions, that were experimentally varied to help frame the experimental data used to identify the iModulons for the reader upfront.

We thank the reviewer for the helpful suggestion. We have included Table 1. on p. 18, which provides a brief description of the samples in the context of each project (reference, condition, bacterial strain, host genotype, infiltration agents). We have also directed readers to Supplementary Data S1, which provides a more detailed description of the samples used in the study at p. 3 lines 102-104: "These samples have been briefly described in Table 1, with a more detailed description included in Supplementary Data S1."

2. p. 4, last paragraph: The authors discuss the downregulation of the flagella iModulons in the context of efflux pumps and antibiotic resistance. There is no discussion in the manuscript about biofilm formation in *P. syringae*. Motility and biofilm formation are often inversely regulated. Were the biofilm genes not within an iModulon and therefore not part of the analysis? This raises the question, are there key physiological outputs that were not captured within iModulons (i.e., what is missed)? What about the 36% of gene expression not included (p. 7 second to last line). Some discussion of the limitations of the iModulon approach seems important to include.

Not all biofilm-associated genes were within iModulons. For example, regulatory genes such as *algU* and those associated with alginate biosynthesis were found within iModulons, whereas genes such as the *wss* operon genes did not belong to any iModulon. This may be a result of the conditions under which the samples included in this study were generated. These conditions may not elicit strong variation in the expression of certain genes, such as biofilm-associated genes. Due to the dependence of the iModulons identified on the condition space, there is a possibility of key physiological outputs not being captured by the iModulons. The tradeoff you mention between motility and biofilm has been observed in other iModulon structures with biofilm-activating conditions, such as that for *Bacillus subtilis* (Rychel et al; PMID: 33311500).

The 64% of variance in the dataset that the iModulons capture refers to the variance in gene expression that is captured by reconstructing the dataset using only iModulon information. The remaining 36% of the variance in gene expression is not captured by the ICA algorithm because it does not exhibit coherent co-regulation and is therefore generally assumed to be noise. The inclusion of more data from diverse conditions in future studies may help the signal:noise ratio that iModulons can capture. We have added the following at p. 9 lines 366-375 to discuss the limitations of iModulon analysis: “The ICA algorithm used to identify iModulons assumes that the activity of each regulator is statistically independent from one another, has a non-Gaussian distribution, and effects genes in a linear manner(3). While these assumptions allow ICA to outcompete other algorithms in quantitatively representing the TRN, more complex regulatory mechanisms may not be captured(42). Additionally, our decomposition of the *P. syringae* TRN does not explain 36% of the variance in the dataset. This is due to a lack of clear co-regulation, and is generally assumed to be noise. Future research can work towards a more comprehensive decomposition of the *P. syringae* TRN through the inclusion of more data(5,43). Appending additional data generated under diverse conditions to the current dataset may result in a change in the set of iModulons identified by ICA, such as new iModulons being identified, large multi-purpose iModulons splitting into iModulons representing each functionality individually, and reduction of noise.” with citations to Saelens et al 2018 (PMID: 29545622), Sastry et al 2019 (PMID: 31797920), Rychel et al 2020 (PMID: 33311500), and Sastry et al 2021 (PMID: 33529205).

3. p. 5, end of second full paragraph: The text states that "production of tetrahydrofolate was upregulated in all four conditions" and that "tetrahydrofolate plays an important role in purine and pyrimidine synthesis". However, the text then goes on to say that "nucleotide synthesis DECREASES". These statements seem contradictory and it is recommended that the text be rewritten here.

We apologize for the unclear wording of the text. We have revised the text to indicate that as a result of the competition strategies of *B. thailandensis*, nucleotide synthesis is upregulated to a lower extent in cocultures containing *B. thailandensis* in comparison to that in its absence. (p. 6 lines 220-223)

4. The references need to be further proofed for consistent use of italics for organism names and correct use of capital letters in titles (only the first word).

We have updated the reference style for the main manuscript as well as that of the Supplementary Material to match that of ASM, and have ensured that only the first word of the title is capitalized and all organism names are italicized. (p. 12-16; Supplemental Material p. 3,4)

Reviewer #2:

Comments on the manuscript “Machine learning uncovers the *Pseudomonas syringae* transcriptome in microbial communities and during infection”

In the manuscript “Machine learning uncovers the *Pseudomonas syringae* transcriptome in microbial communities and during infection”, Bajpe and colleagues use publicly available RNA seq datasets to describe the transcriptome of *P. syringae* upon interaction with other bacterial species and during infection of *Arabidopsis thaliana*, and map transcriptional networks. This study provides the first thorough analysis of gene regulation occurring when *P. syringae* interacts with other microbes and infects plants, thereby advancing the basic knowledge of how the bacteria reprograms its gene expression in response to environmental cues typically encountered during its lifestyle, and how this benefits its interaction with other bacteria and plants. Such analysis will provide key resources for researchers willing to investigate the molecular basis of some of the transcriptional regulations described in the paper. The manuscript is very well written and scientifically sound. I have very few comments, most of them related to the resolution of the figures, which should be increased, and to data interpretation.

Comments

1) The resolution of the figures should be increased as most of them appear pixelated on the printed version of the manuscript or upon zooming.

We apologize for the inconvenience. The figures have been resubmitted as EPS files with a resolution of 300 ppi.

2) Supplementary Note S1. The authors claim that *C. violaceum* was found to upregulate the activities of the Resistance-1 and Resistance-2 iModulons in *P. syringae*. However, according to the heatmap in Figure 2a, only Resistance-2 iModulon, and not Resistance-1, is upregulated when *P. syringae* interacts with *C. violaceum*. The authors should change or further clarify their data interpretation in the text.

We apologize for the unclear wording of the text. The Resistance-1 iModulon was upregulated in cocultures containing *C. violaceum* in comparison to the monoculture. Considering that the extent of this upregulation was not as high as that of the Resistance-2 iModulon, we have modified the text to add that only the Resistance-2 iModulon was notably upregulated. (Supplemental Material p. 1, lines 7-8)

3) Results. Lines 174-186. While the decrease of expression of siderophore genes in the presence of *B. thailandensis* may reflect a strategy adopted by *P. syringae* for decreasing siderophore production to prevent piracy, as suggested by the authors, it could also indicate that *P. syringae* somehow experiences iron sufficiency when cocultured with *B. thailandensis*, therefore promoting Fur-mediated transcriptional repression of siderophore genes. Iron sufficiency, and especially overabundance, can also promote the formation of free radicals

through the Fenton reaction, which could also explain the induction of the FeS/Oxidative Stress iModulon in *P. syringae* upon coculture with *B. thailandensis*. The authors should maybe consider evoking these possibilities in the text.

This is a very interesting observation. We have added the following: “Alternatively, the downregulation of both siderophore-associated iModulons may indicate that *P. syringae* experiences iron sufficiency in the presence of *B. thailandensis*. This would result in Fur-mediated downregulation of siderophore biosynthesis(27). It is notable that the Fe-S/Oxidative Stress iModulon is upregulated in cocultures containing *B. thailandensis*, as an abundance of iron can contribute to the production of reactive oxygen species (ROS) through the Fenton reaction.”, including a citation to Butcher et al, 2011 (PMID: PMC3165696). (p. 5, lines 205-209)

Thank you for the thoughtful addition to this section!

4) Results. Lines 188-199. The authors suggest that the induction of the Recombination iModulon may capture the response of the bacteria to DNA damages. One may consider the possibility of these DNA damages to be the result of oxidative stress experienced by *P. syringae* when cocultured with *B. thailandensis*, as suggested by the induction of the Fe-S/Oxidative Stress regulon. The authors should discuss this possibility in the text.

This is another interesting connection, thank you. We have added the following: “The upregulation of the Recombination iModulon may also be due to oxidative stress, as evidenced by the upregulation of the Fe-S/Oxidative Stress iModulon in coculture conditions.” (p. 6, lines 215-217)

Minor comments

1) Results. Line 142. “[...] RND and MFS efflux pumps [...]”. Authors should spell out the RND (Resistance-Nodulation- Division) and MFS (Major Facilitator Superfamily) abbreviations the first time they appear in the text.

We have added the full form of the abbreviations in the text. (p. 5, lines 161-162)

2) Figure 2a. The Ps, PsCv, PsBt, and PsCvBt abbreviations should be spelled out in the figure legend.

We have replaced the definition of the abbreviations for ‘Ps’, ‘Bt’, and ‘Cv’ with definitions for ‘Ps’, ‘PsCv’, ‘PsBt’, and ‘PsCvBt’. (p. 17, lines 659-660)

3) Supplementary Note S1. “*C. violacium*”. Replace “violacium” with “violaceum”.

Thank you for the close reading and catching this error. This has been revised to correct the spelling (Supplemental Material p. 1, line 7)

July 19, 2023

Prof. Bernhard O Palsson
UCSD
San Diego

Re: mSystems00437-23R1 (Machine learning uncovers the *Pseudomonas syringae* transcriptome in microbial communities and during infection)

Dear Prof. Bernhard O Palsson:

Your manuscript has been accepted, and I am forwarding it to the ASM Journals Department for publication. For your reference, ASM Journals' address is given below. Before it can be scheduled for publication, your manuscript will be checked by the mSystems production staff to make sure that all elements meet the technical requirements for publication. They will contact you if anything needs to be revised before copyediting and production can begin. Otherwise, you will be notified when your proofs are ready to be viewed.

If you would like to submit a potential Featured Image, please email a file and a short legend to msystems@asmusa.org. Please note that we can only consider images that (i) the authors created or own and (ii) have not been previously published. By submitting, you agree that the image can be used under the same terms as the published article. File requirements: square dimensions (4" x 4"), 300 dpi resolution, RGB colorspace, TIF file format.

We recognize that the video files can become quite large, and so to avoid quality loss ASM suggests sending the video file via <https://www.wetransfer.com/>. When you have a final version of the video and the still ready to share, please send it to mSystems staff at msystems@asmusa.org.

Sincerely,

Ryan McClure
Editor, mSystems

Journals Department
E-mail: mSystems@asmusa.org